# HDAC Inhibitor Abrogates LTA−Induced PAI-1 Expression in Pleural Mesothelial Cells and Attenuates Experimental Pleural Fibrosis

**DOI:** 10.3390/ph14060585

**Published:** 2021-06-18

**Authors:** Wei-Lin Chen, Mei-Chuan Chen, Shang-Fu Hsu, Shih-Hsin Hsiao, Chi-Li Chung

**Affiliations:** 1Department of Nursing, MacKay Junior College of Medicine, Nursing, and Management, Taipei 112, Taiwan; s519@mail.mkc.edu.tw; 2Division of Pulmonary Medicine, Department of Internal Medicine, Taipei Medical University Hospital, Taipei 110, Taiwan; calmwave@gmail.com (M.-C.C.); hsushangfu@gmail.com (S.-F.H.); 3Division of Pulmonary Medicine, Department of Internal Medicine, School of Medicine, College of Medicine, Taipei Medical University, Taipei 110, Taiwan; 4School of Respiratory Therapy, College of Medicine, Taipei Medical University, Taipei 110, Taiwan

**Keywords:** histone deacetylase inhibitor, lipoteichoic acid, plasminogen activator inhibitor-1, pleural fibrosis, pleural mesothelial cell, residual pleural thickening

## Abstract

Lipoteichoic acid (LTA) stimulates pleural mesothelial cell (PMC) to overproduce plasminogen activator inhibitor-1 (PAI-1), and thus may promote pleural fibrosis in Gram-positive bacteria (GPB) parapneumonic effusion (PPE). Histone deacetylase inhibitor (HDACi) was found to possess anti-fibrotic properties. However, the effects of HDACi on pleural fibrosis remain unclear. The effusion PAI-1 was measured among 64 patients with GPB PPE. Pleural fibrosis was measured as radiographical residual pleural thickening (RPT) and opacity at a 12-month follow-up. The LTA−stimulated human PMCs and intrapleural doxycycline−injected rats were pretreated with or without the pan-HDACi, m-carboxycinnamic acid bis-hydroxamide (CBHA), then PAI-1 and collagen expression and activated signalings in PMCs, and morphologic pleural changes in rats were measured. Effusion PAI-1 levels were significantly higher in GPB PPE patients with RPT > 10 mm (*n* = 26) than those without (*n* = 38), and had positive correlation with pleural fibrosis shadowing. CBHA significantly reduced LTA−induced PAI-1 and collagen expression via inhibition of JNK, and decreased PAI-1 promoter activity and mRNA levels in PMCs. Furthermore, in doxycycline−treated rats, CBHA substantially repressed PAI-1 and collagen synthesis in pleural mesothelium and minimized pleural fibrosis. Conclusively, CBHA abrogates LTA−induced PAI-1 and collagen expression in PMCs and attenuates experimental pleural fibrosis. PAI-1 inhibition by HDACi may confer potential therapy for pleural fibrosis.

## 1. Introduction

Parapneumonic pleural effusion (PPE) is a common sequel of pneumonia where bacteria invade the pleural cavity and stimulate pleural mesothelial cells (PMCs) to induce inflammation and coagulation [1], characterized by inflammatory cytokine surge, plasminogen activator inhibitor-1 (PAI-1) overproduction, fibrin deposition and collagen accumulation that often leads to pleural loculation and fibrosis and a significant morbidity and mortality [2]. Excess PAI-1 inactivates plasminogen activators, inhibits fibrinolysis and extracellular matrix proteolysis, and alters tissue remodeling [3]. Previous reports demonstrated that elevated PAI-1 is significantly associated with residual pleural thickening (RPT) in PPE [4,5], suggesting that PAI-1 may play a key role in development of pleural fibrosis [3].

Gram-positive bacteria (GPB), such as *Streptococcus* or *Staphylococcus* species are common pathogens of PPE [6]. Lipoteichoic acid (LTA), an outer membrane component of GPB, attaches to host cells to initiate inflammation and enhance coagulation [7], and may predispose to pleural fibrosis [8]. Moreover, our recent study demonstrated that GPB PPE has remarkably higher levels of PAI-1 than culture-negative PPE and that LTA noticeably increases PAI-1 synthesis by human PMCs [9], implying that LTA stimulation of PMCs may give rise to fibrotic sequel of GPB PPE via induction of PAI-1. 

Histone deacetylase inhibitor (HDACi) has been reported to correct aberrant gene expression and attenuate various organ fibrosis in vivo [10]. However, the therapeutic role of HDACi in pleural fibrosis remains to be elucidated. Our group previously showed that the pan-HDACi, m-carboxycinnamic acid bis-hydroxamide (CBHA), significantly reduced PAI-1 expression in human PMCs triggered by transforming growth factor-β1 (TGF-β1) and tumor necrosis factor-α (TNF-α) [11,12]. Conversely, another pan-HDACi trichostatin A (TSA) paradoxically amplified lipopolysaccharide−induced PAI-1 expression in bone marrow−derived macrophages [13], which therefore merits further studies on the modulation of HDACi on LTA-triggered PAI-1 production. 

Altogether, all the abovementioned preclinical data suggest that LTA−mediated PAI-1 overproduction may be substantially implicated in the pleural fibrogenesis and that HDACi may present as a promising treatment option for tissue fibrosis. Nonetheless, to our knowledge, the in vitro and in vivo effects of HDACi on PAI-1 expression in pleural infection and its residual fibrosis has rarely been investigated. Therefore, the aims of this study were to delineate the profibrotic clinical significance of PAI-1 in GPB PPE, the regulation of CBHA on LTA−driven PAI-1 and collagen expression in human PMCs, and the therapeutic effects of CBHA on experimental pleural fibrosis. 

## 2. Results

### 2.1. Pleural Fluid Pamameters between Gram-Positive Bacteria (GPB) Parapneumonic Pleural Effusion (PPE) Patients with Residual Pleural Thickening (RPT) ≤ 10 mm and RPT > 10 mm

As shown in Table 1, consecutive 64 patients with GPB PPE were recruited, including 42 men and 22 women with an age range from 42 to 80 years. There were 27 patients presenting with *Streptococcus* species (*S. pneumoniae* in 20, *S. anginosus* in 5, *S. constellatus* in 2) and 37 with *Staphylococcus* species (*S. aureus* in 33, *S. capitis* in 4) in their pleural fluid culture. All patients finished 12 months of follow-up from June 2015 through December 2018. To explore the profibrotic significance of PAI-1, the GPB PPE patients were categorized into RPT ≤ 10 mm (*n* = 38) and RPT > 10 mm (*n* = 26) groups, measured on the 12-month follow-up chest radiograph (CXR). The pleural effusion variables showed that RPT > 10 mm patient group had substantially higher levels of pleural fluid lactate dehydrogenase (LDH) than did those with RPT ≤ 10 mm, while there was no meaningful difference in values of pH, glucose, protein, and leukocyte count between two groups. Additionally, the levels of PAI-1, TNF-α, interleukin (IL)-1β and IL-8 were remarkably elevated in patients with RPT > 10 mm, compared to RPT ≤ 10 mm group. Furthermore, there were significantly larger effusion CXR score on presentation and lower forced vital capacity (FVC) at the end of follow up among the former than the latter group, which may signify more intense inflammation and larger pleural fluid formation in RPT > 10 mm patients. The results indicated that the enhanced inflammation, repressed fibrinolysis and especially the increased PAI-1 are linked to pleural fibrosis among patients with GPB PPE.

### 2.2. Multivariate Logistic Regression Analysis for Variables Related to RPT > 10 mm among GPB PPE Patients

To further ascertain the importance of proinflammatory cytokines and PAI-1 in pleural fibrosis among GPB PPE patients, we used multivariate logistic regression analysis to recognize the factors related to RPT > 10 mm at the end of 12-month follow-up (Table 2). Significant variables in univariate analysis were incorporated for investigation and the results demonstrated that only higher pleural fluid PAI-1 level was associated with RPT > 10 mm in GPB PPE (odds ratio = 2.72, 95% confidence interval (CI) = 1.27–5.84, *p* = 0.006).

### 2.3. Optimal Sensitivity, Specificity, and Cutoff Value of Factors to Predict RPT > 10 mm

Moreover, the receiver operating characteristic (ROC) curves demonstrated that the pleural fluid PAI-1 at the cutoff level > 83.5 pg/mL had greatest sensitivity and specificity to predict RPT > 10 mm among GPB PPE individuals (area under the ROC curve = 0.978, 95% confidence interval (CI) = 0.951–1.000; sensitivity 84.6%, 95% CI = 65.1–95.6%; specificity 97.4%, 95% CI = 86.2–99.9%) (Figure 1A)

### 2.4. Correlation between Effusion Plasminogen Activator Inhibitor-1 (PAI-1) and Residual Pleural Fibrosis

Since pleural fluid PAI-1 level had the value of predicting RPT > 10 mm among the GPB PPE patients (Figure 1A), we evaluated its link with area of residual pleural fibrosis. As shown in Figure 1B, the levels of PAI-1 correlated positively with the pleural shadowing CXR score at the end of 12-month follow-up, indicating that PAI-1 is mainly involved in the fibrogenesis of GPB PPE. Accordingly, it merits further laboratory works to validate the profibrotic role of PAI-1 production by PMCs in GPB PPE and the feasibility of PAI-1 inhibition by HDACi as a potential therapy for pleural fibrosis.

### 2.5. CBHA Inhibits Lipoteichoid Acid (LTA)-Induced PAI-1 and Collagen Expression in Human Pleural Mesothelial Cells (PMCs)

To examine the effect of GPB infection on PAI-1 expression in human PMCs, the cultured PMCs were stimulated with LTA (0.5, 1 and 5 μg/mL) for 24 h, and, with the maximal effect at 1 μg/mL, LTA significantly stimulated the production of PAI-1, as compared with the resting state (Figure 2A). In line with our previous report [9], there was no dose-response at the highest concentration of 5 μg/mL, which may suggest a ceiling effect of LTA on PAI-1 expression in PMCs. Therefore, 1 μg/mL of LTA was selected as the optimal dose for subsequent experiments. To examine the influence of CBHA on LTA−mediated PAI-1 synthesis, PMCs were treated with DMSO (vehicle) or CBHA for 15 min, followed by LTA for 24 h. It showed that CBHA abrogated the stimulatory effect of LTA on PAI-1 production (Figure 2B). In parallel, CBHA concentration-dependently inhibited LTA-induced collagen production in PMCs, compared to the vehicle group (Figure 2C). These results imply that CBHA may ablate LTA-mediated fibrin formation and fibrogenesis in the pleural space.

### 2.6. CBHA Blocks LTA-Induced JNK Phosphorylation in Human PMCs

Furthermore, we used specific pharmacologic inhibitors of MAPKs, PI3K/AKT or NF-κB to examine the signaling pathways of PAI-1 production induced by LTA and the result showed that LTA-mediated PAI-1 synthesis was significantly minimized by pretreatment with SP600125 (JNK inhibitor), but not by Parthenolide (NF-κB inhibitor), LY294002 (PI3K inhibitor), PD98059 (MEK inhibitor), nor SB203580 (p38 MAPK inhibitor) in PMCs (Figure 3A). Comparably, the JNK inhibitor (SP600125) significantly repressed LTA-induced collagen expression, compared with vehicle (Figure 3B). Consistently, as compared to the resting state, LTA notably stimulated JNK phosphorylation, with a maximum response at 15 min (Figure 3C). Accordingly, to verify the effect of CBHA on LTA-activated signaling mediating the PAI-1 and collagen I expression, PMCs were treated with CBHA or DMSO (vehicle), followed by LTA stimulation and it revealed that LTA-activated JNK phosphorylation was substantially ablated by CBHA (Figure 3D). In addition, to further validate the role of JNK in LTA-induced PAI-1 expression, we used small interfering RNA (siRNA) against the 3 genes encoding JNK protein, including JNK1, JNK2, and JNK3, to silence JNK expression [14]. As shown in Figure 3E, LTA upregulated JNK and PAI-1 expression (lane 3), while partial silencing of JNK by JNK3 siRNA, but not by JNK1 or JNK2 (data not shown), notably attenuated LTA-stimulated PAI-1 expression (lane 4), signifying the role of JNK3 in LTA-activated pathway. Moreover, knockdown of JNK3 did not affect the inhibitory effect of CBHA on LTA-mediated induction of PAI-1 (Figure 3E, lane 6), suggesting that CBHA may abrogate LTA-induced PAI-1 through disruption of various cellular processes, including but not limited to JNK signaling.

### 2.7. CBHA Reduces LTA-Induced PAI-1 mRNA Expression and Promoter Activity in Human PMCs

To further investigate whether CBHA reduced LTA-stimulated PAI-1 protein translation via repression of PAI-1 gene transcription, RT-PCR analysis was performed. Compared to the resting state, LTA substantially upregulated PAI-1 mRNA expression at 6 h, while pretreatment with various concentrations of CBHA markedly suppressed the increased PAI-1 mRNA levels (Figure 4A). Furthermore, the luciferase reporter assay revealed that CBHA concentration-dependently mitigated the PAI-1 promoter activity stimulated by LTA (Figure 4B).

### 2.8. CBHA Minimizes Doxycycline–Induced PAI-1 and Collagen Expression on Visceral Pleural Mesothelium and Pleural Fibrosis in Rat Model

Accordingly, based on the above in vitro results, we used doxycycline pleural fibrosis rat model to further delineate the in vivo antifibrotic activity of CBHA [15]. We investigated whether CBHA could attenuate the PAI-1 expression and fibrosis in the visceral pleura of rats treated with intrapleural injection of doxycycline. The histological examination revealed a monolayer of PMCs on the visceral pleural surface of the rats in the control group (Figure 5A). Compared with control, the doxycycline (10 mg/Kg)–treated group exhibited significant fibrous thickening in the visceral pleura (Figure 5B). However, pretreatment with CBHA markedly suppressed these fibrotic changes induced by doxycycline (Figure 5C). The mean pleural thickness measured was substantially greater in the doxycycline group than in the control group, and CBHA distinctly attenuated the induced pleural thickening (Figure 5J). Additionally, the intensity of PAI-1 immunostaining in the visceral pleura was notably increased in the doxycycline-treated rats compared with the control group (Figure 5D,E). Contrarily, pretreatment with CBHA substantially decreased the doxycycline-induced PAI-1 expression (Figure 5F). Moreover, the western blot analyses of the resected visceral pleura revealed that, as compared to the control, PAI-1 expression was notably upregulated in the doxycycline–treated group, whereas CBHA markedly suppressed this inducing effect (Figure 5K). Furthermore, Masson’s trichrome staining showed that doxycycline markedly induced collagen deposition in the visceral pleura, which was considerably attenuated by pretreatment with CBHA (Figure 5G–I). The measured percentage of collagen deposition area in the visceral pleura increased significantly after doxycycline instillation, as compared with the control. In contrast, CBHA remarkably reduced the doxycycline–induced collagen accumulation (Figure 5L). These results demonstrated that CBHA effectively minimize doxycycline–induced PAI-1 and collagen expression as well as pleural fibrosis in the visceral pleurae of rats.

Collectively, all these findings revealed that GPB infection triggers pleural injury via LTA, elicits PAI-1 and collagen production, and thus give rise to pleural fibrosis. Alternatively, CBHA significantly reduced LTA−induced PAI-1 expression via repression of JNK and PAI-1 promoter activity and decreased collagen I production by PMCs. Furthermore, in doxycycline−treated rats, CBHA reduced PAI-1 expression and collagen deposition in pleural mesothelium and effectively attenuated pleural fibrosis (Figure 6). Further experiments are needed to entirely depict the CBHA-modulated signal pathways in PMCs.

## 3. Discussion

The current study revealed that the pleural fluid levels of PAI-1 were remarkably higher among GPB PPE patients with meaningful residual pleural fibrosis (RPT > 10 mm) than those without (RPT ≤ 10 mm). Moreover, effusion PAI-1 correlated positively with pleural fibrosis area and was an independent predictor for RPT > 10 mm in GPB PPE. In human PMCs, LTA notably increased the expression of PAI-1 and collagen I. Instead, CBHA significantly reduced PAI-1 expression through inhibition of JNK phosphorylation and PAI-1 promoter activity, and decreased collagen I production by PMCs upon LTA stimulation. Furthermore, in doxycycline−treated rats, CBHA minimized PAI-1 expression and collagen deposition in the pleural mesothelium and effectively diminished pleural fibrosis. To the best of our knowledge, this is the first research to denote the clinical importance and profibrotic implication of LTA−activated upregulation of PAI-1 in GPB PPE and to clarify the antifibrotic effect and therapeutic potential of CBHA on pleural fibrosis.

PAI-1 can repress fibrinolysis and matrix degradation and is essential for tissue fibrosis [3]. Prior studies revealed that effusion PAI-1 level was notably higher among PPE patients with RPT than those without [4,5], and substantially greater in culture positive PPE, particularly GPB PPE, than culture negative PPE [9], indicating that PAI-1 is implicated in the fibrotic sequel of PPE. Accordingly, the present study aims to verify the fibrogenic clinical significance of PAI-1 in patients with GPB PPE. We demonstrated that among the 64 GPB PPE patients, the effusion PAI-1 level positively correlated with the area of residual fibrosis and may serve to be a predictor for the clinically significant parameter of RPT > 10 mm, which imply that PAI-1 is an essential player in pleural fibrosis in GPB PPE and targeting PA-1 expression may be a potential treatment option for pleural fibrosis.

Our recent study has identified that LTA, the important cell wall polymer of GPB, significantly elicits PAI-1 expression in PMCs via TLR2/JNK/AP-1 signal pathway [9], suggesting that therapies directing against LTA-mediated PAI-1 production may attenuate fibrotic sequels of GPB PPE. HDACs are essentially implicated in tissue inflammation and fibrosis and HDACi are extensively investigated in fibrosis research because they ablate inflammation, fibrinolysis, and epithelial-mesenchymal transition (EMT) [10]. Previous studies showed that the pan-HDACi TSA repressed the synthesis of collagen I and III in rat hepatic stellate cells [16], and that both structure-related pan-HDAC inhibitors CBHA and suberoylanilide hydroxamic acid (SAHA) have been shown to target HDCA1 and HDAC3 [17], and attenuated various organ fibrosis [10]. Accordingly, the present study explored the actions of HDACi on pleural fibrosis by the use of CBHA. Besides, we formerly reported that CBHA effectively abrogates TGF-β1− or TNF-α−mediated PAI-1 synthesis in PMCs [11,12]. Concurring with prior reports [11,12,16], the current study verified that LTA stimulated PAI-1 expression at both the protein and mRNA levels, and this inducing effect was distinctly diminished by CBHA. Moreover, our data showed that LTA concurrently induced the expression of collagen, the most notable marker of fibrosis, which was also significantly inhibited by CBHA. Further experiments using HDACi with similar actions, such as SAHA, and studies of the HDAC inhibition effects on LTA-mediated myofibroblast transformation in PMCs are warranted for more clarification of the pleural fibrogenesis in GPB PPE.

HDACi regulation on gene expression can occur at multiple levels of cellular processes, including signal transduction, gene transcription, mRNA steadiness to protein decay [18]. A recent report showed that TSA might target JNK−dependent Notch-2 signaling to attenuate renal fibrosis [19]. Consistent with our previous report [9], LTA−induced PAI-1 and collagen I expression were markedly suppressed by JNK inhibitor (SP100625), and pretreatment with CBHA markedly decreased LTA−activated JNK phosphorylation and subsequently repressed PAI-1 transcription activity and gene expression in human PMCs. These results feature that JNK pathway inhibition underlies the suppression of LTA−induced PAI-1 and collagen I production by CBHA.

Subsequently, given the downregulation impact of CBHA on LTA−activated PAI-1 and collagen I synthesis, HDACi should be examined in vivo as a promising treatment for pleural fibrosis [20]. The present study translated the in vitro data into in vivo experiments using a verified animal model of pleural fibrosis [15], and revealed that a single application of CBHA significantly reduced doxycycline-induced pleural fibrous thickness and collagen accumulation in the pleura. Since dysregulation of PAI-1 expression leads to tissue fibrosis [2], our findings additionally exhibited that PAI-1 expression was increased in doxycycline−stimulated pleural mesothelium and CBHA markedly attenuated this inductive effect. All these findings provided the in vivo evidence of abrogation effect of CBHA on PAI-1 synthesis and fibrogenesis and indicated that HDAC inhibition may be used as a strategy to deter the development or progression of pleural fibrosis.

However, some limitations existed in this study. First, given the ability of HDACi to modulate signalings in various cellular processes [18], the mechanism underlying CBHA inhibition on LTA−induced PAI-1 and collagen I expression may not be limited to repression of JNK phosphorylation only. Further experiments exploring the regulation of CBHA on various nonhistone proteins engaged in transcription and post-transcriptional processes are needed, as depicted in our previous report [11]. Second, we used doxycycline rather than LTA as the pleural fibrosing agent because of the inadequacy of LTA alone to produce remarkable fibrosis. Besides, developing a model mimicking PPE by direct inoculation with common gram-positive bacteria has been unsuccessful, as the animals either died of sepsis or did not develop eminent pleural infection [21,22]. Doxycycline effectively causes chemical pleural irritation to develop fibrosis whereas the fibrogenesis in PPE is more complex. The similarities between doxycycline- and LTA-induced pleural fibrosis are that both agents trigger pleural inflammation to elicit PAI-1 production to inhibit fibrinolysis. Instead, the divergences are that doxycycline, like most other pleural fibrosing agents, may cause direct injury to PMCs to elaborate TNF-α and TGF-β1, and thereby induce PAI-1 expression [23]; while LTA specifically binds to toll-like receptor 2 (TLR2) on PMCs to activate JNK signaling and stimulate PAI-1 synthesis [9]. Accordingly, doxycycline may generate pleural fibrosis by overproducing PAI-1 via distinct cytokine-driven pathways that, as demonstrated in previous reports [11,12], may also be abrogated by HDACi. Although the doxycycline model may not exactly emulate the human postinfectious pleural sequel, to our best knowledge, this is the premiere animal research to verify the therapeutic action of PAI-1 and collagen inhibition by HDACi on pleural fibrosis. Further human trials are mandated to validate whether CBHA can be used safely to attenuate infection-related pleural fibrosis and its long-term effectiveness.

## 4. Materials and Methods

### 4.1. Materials

LTA from *staphylococcus aureus,* CBHA, doxycycline and all other reagents were procured from Sigma-Aldrich (St. Louis, MO, USA). All antibodies were obtained from Abcam (Cambridge, MA, USA), except that for PAI-1 (BD Biosciences, San Jose, CA, USA). LY294002, parthenolide, SB203580, SP600125, PD98059 were acquired from MedChemExpress (Monmouth Junction, NJ, USA).

### 4.2. Patient Enrollment

Serial patients who had pleural effusions and were admitted to Taipei Medical University Hospital, underwent thoracentesis and were enrolled when the diagnosis of GPB PPE was established based on the pleural fluid bacteria culture. Ethics approval TMU-JIRB No. 201504078 was acquired from the Institutional Review Board of Taipei Medical University Hospital (Taipei, Taiwan). All patients offered signed document of informed consent before enrollment. Patients with records of bleeding diathesis, anticoagulant therapy and invasive pleural procedures were excluded.

### 4.3. Thoracentesis and Analysis of Pleural Fluids

With the guidance of chest ultrasonography, 50 mL of pleural effusion was obtained by needle aspiration. Pleural fluid analyses were done as routine and 5 mL of pleural fluid was injected into aerobic and anaerobic culture bottles, respectively. Additionally, two pleural fluid samples were inoculated into the matching blood culture bottles to enhance the bacteria growth yield [24]. Standardized broad-spectrum antibiotics were prescribed initially that were later adjusted based on the culture result.

### 4.4. Chest Radiographs (CXRs) and Pulmonary Function

Chest radiographs (CXRs) were acquired on admission, every 2 months during follow-up, and at the end of 12 months. The picture archiving and communication system (PACS) was used to store CXRs as digital images and the pleural shadowing and the hemithorax areas were calculated by an image–processing program. Two independent radiologists blinded to clinical information read each CXR to determine (a) RPT: the largest linear width of pleural shadowing and (b) CXR score of the area of pleural effusion or thickening [25]. Clinically meaningful RPT was quantified as a lateral pleural thickening of >10 mm at the completion of 12-month follow-up [26]. All patients received pulmonary function tests at the end of follow-up period.

### 4.5. Measurement of Cytokines and Fibrinolytic Factors

The commercially available enzyme–linked immunosorbent assay (ELISA) kits were employed to measure the pleural fluid levels of TNF-α, IL-1β, IL-8, PAI-1, and tPA (R & D System; Minneapolis, MN, USA) as formerly depicted [25].

### 4.6. Human Pleural Mesothelial Cell Culture

The primary human PMCs were cultivated from pleural effusions of heart failure patients and MeT-5A pleural mesothelial cell line was procured from American Type Culture Collection (ATCC^®^ CRL-9444™; ATCC, Manassas, VA, USA). Both kind of PMCs cells were incubated and maintained, as previously described [27].

### 4.7. Western Blotting Assay

The proteins in total cell lysates were separated by electrophoresis on SDS-PAGE gel and transferred to a PVDF membrane. Subsequently, the membranes were probed with specific antibodies against various proteins of interest, and washed three times for 5 min and then incubated with the HRP-conjugated secondary antibody for 1 h. The densitometric quantification was conducted, as formerly reported [25].

### 4.8. RNA Interference

MeT-5A cells were transfected with a control siRNA or a siRNA against JNK1, JNK 2 or JNK3, using the DharmaFECT^®^ siRNA transfection reagent (Thermo Scientific; Waltham, MA, USA). The transfection reagent containing siRNA was added to PMCs in serum-free media for 24 h. The media was changed to fresh serum-free M199 for another 24 h then treated with LTA with or without CBHA pretreatment to examine JNK and PAI-1 proteins, respectively [9].

### 4.9. RNA Extraction and Reverse Transcription-Polymerase Chain Reaction (RT-PCR)

After treatment, total RNA was extracted using the TRIsure^®^ reagent (Bioline; London, UK), and 1 μg RNA was submitted for cDNA synthesis (Super Script On-Step RT-PCR system, Thermo Fisher Scientific; Waltham, MA, USA). PCR products were fixed on agarose gels and stained with ethidium bromide. Specific primer sequences (sense/antisense) were formulated as below: PAI-1: 5′-TGCTGGTGAATGCCCTCTACT-3′/5′-CGGTCATTCCCAGGTTCTCTA-3′; GAPDH: 5′-GCCGCCTGGTCACCAGGGCTG-3′/5′-ATGGACTGTGGTCATGAGCCC-3′.

### 4.10. PAI-1 Luciferase Activity Assay

MeT-5A cells were co-transfected with PAI-1 and Renilla reporter plasmids, using the TurboFect transfection reagent (Thermo Fisher Scientific; Waltham, MA, USA). After 24 h incubation, the luciferase activity was measured after cell lysis as formerly depicted [11].

### 4.11. Pleural Fibrosis Rat Model

The animal research was conducted following the National Research Council Guide for the Care and use of Laboratory Animals, and was approved by the Institutional Animal Care and Use Committee (IACUC No.: LAC-99-0212) of Taipei Medical University. Male Wistar rats (200–250 g) were intraperitoneally anesthetized with 400 mg/Kg of choloral hydrate. Intrapleural injection was started with a longitudinal skin incision about 2 cm in the right parasternal area, approximately 2 cm above the costal margin, and the parietal pleura was exposed after blunt dissection of chest wall muscles [28]. The reagents were slowly pushed into the right pleural cavity with a 1 mL syringe, then muscles and skin were stitched with 5‒0 nylon.

All animals were randomly allocated into three groups: (A) Doxycycline group: injected intrapleurally with doxycycline (10 mg/Kg); (B) Doxycycline with CBHA group: injected intrapleurally with a single dose of CBHA (100 mg/Kg) for 15 min followed by intrapleural doxycycline treatment; and (C) Control group: injected intrapleurally with equal volume of vehicle (PBS). The dose for CBHA was determined based on pilot experiments directed to establish the effects of different doses on the thickness of the submesothelial area [29]. At a dose of 50 mg/kg, CBHA minimally decreased the pleural thickness induced by injected-doxycycline. Therefore, we selected CBHA (100 mg/kg) in the present study. CBHA was dissolved in the co-solvent of alcohol, cremophor and PBS solution. On day 7, the rats were sacrificed by asphyxiation with CO2. The pleural tissue was resected and collected as described previously [30].

### 4.12. Histology and Immunohistochemistry

Visceral pleura sections from the right hemithorax were stained by hematoxylin and eosin (H&E) and the microscopic fibrosis was evaluated. The pleural thickness was scored using the Leica Q500IW Imaging Workstation, Processing and Analysis System (Leica Ltd.; Cambridge, UK). Ten different points on each sample in 20× fields were selected for measurement and the results were reported as mean. To assess PAI-1 immunostaining, paraffin-embedded pleural tissues were deparaffinized. Sections (5 μm) were incubated with PAI-1-specific rabbit polyclonal antibody overnight and antibody binding was detected with the use of biotinylated anti-rabbit antibody and horseradish peroxidase streptavidin. Visualization was done with 3,3-diaminobenzidine (Dako; Carpinteria, CA, USA). Images were visualized using Olympus microscope (BX41, Tokyo, Japan). To verify the immunohistochemistry findings, the PAI-1 protein in pleural tissue homogenates was analyzed by Western blot. Additionally, the pleura slices were processed for Masson’s trichrome stain. Areas of fibrosis were analyzed using image pro-express software (Media Cybernetics; Rockville, Maryland, USA). In each pleura section, 10 random points were analyzed. The total collagen percentage was calculated as collagen positive area/total area of the visceral pleura multiplied by 100.

### 4.13. Statistical Analysis

Data were expressed as median (range) or mean ± SEM. Mann–Whitney U test or unpaired t-test was employed for comparisons between two groups, when appropriate. Spearman rank correlation was used to measure the degree of association between nonparametric variables. Fisher’s exact test was used to examine categorical variables between two groups.

Multivariate logistic regression analysis was employed to recognize factors most closely related to development of RPT > 10 mm. Receiver operating characteristics (ROC) curve analysis was used to identify the optimal sensitivity, specificity, and cutoff value of pleural fluid variables to predict RPT > 10 mm. A two-tailed *p*-value < 0.05 was considered to be statistically significant.

## 5. Conclusions

In conclusion, HDACi CBHA abrogates LTA–induced PAI-1 and collagen expression in PMCs and attenuates experimental pleural fibrosis. PAI-1 inhibition by HDACi may confer potential therapy for pleural fibrosis.

## Figures and Tables

**Figure 1 pharmaceuticals-14-00585-f001:**
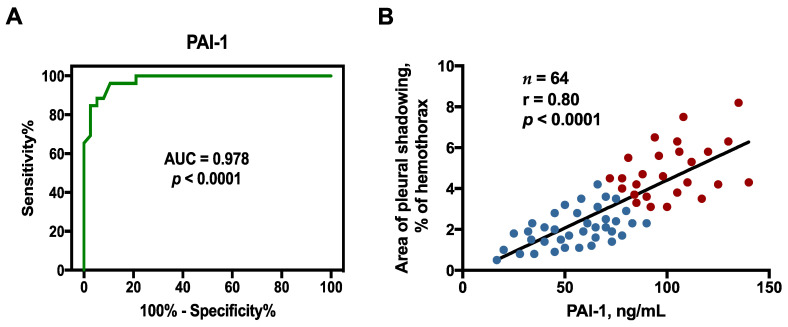
Correlation between pleural fluid PAI-1 level and residual pleural fibrosis among GPB PPE patients: (**A**) ROC curve for PAI-1 value to predict RPT > 10 mm; (**B**) Correlation between PAI-1 and residual pleural opacity CXR score. ROC, receiver operating characteristic; AUC, area under the ROC curve; GPB, Gram-positive bacteria; PPE, parapneumonic pleural effusion. Blue dot, GPB PPE with RPT ≤ 10 mm (*n* = 38); Red dot, GPB PPE with RPT > 10 mm (*n* = 26).

**Figure 2 pharmaceuticals-14-00585-f002:**
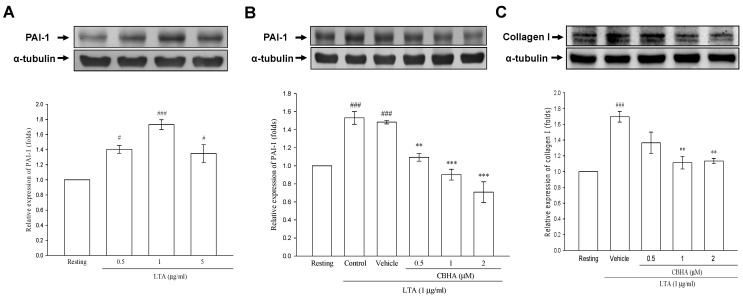
CBHA inhibits LTA-induced PAI-1 and collagen expression in human PMCs. (**A**) Primary pleural mesothelial cells were treated with LTA (0.5–5 μg/mL) for 24 h, and PAI-1 expression was analyzed by western blot and measured by optical densitometry analysis. Data were pooled from four independent experiments and expressed as average fold change (mean ± SEM) in the expression relative to resting group. (**B**,**C**) MeT-5A cells pretreated with various concentrations of CBHA (0.5–2 μM) or vehicle (Dimethyl sulfoxide; DMSO) was stimulated with LTA (1 μg/mL) for 24 h. PAI-1 (**B**) or collagen-I (**C**) expression was analyzed by western blot and measured by optical densitometry analysis. Data were pooled from four independent experiments and expressed as average fold change (mean ± SEM) in the expression relative to resting or vehicle group. **^#^**
*p* < 0.05 and ^###^
*p* < 0.001 compared to the resting group. ** *p* < 0.01 and *** *p* < 0.001 compared to the vehicle (DMSO) group.

**Figure 3 pharmaceuticals-14-00585-f003:**
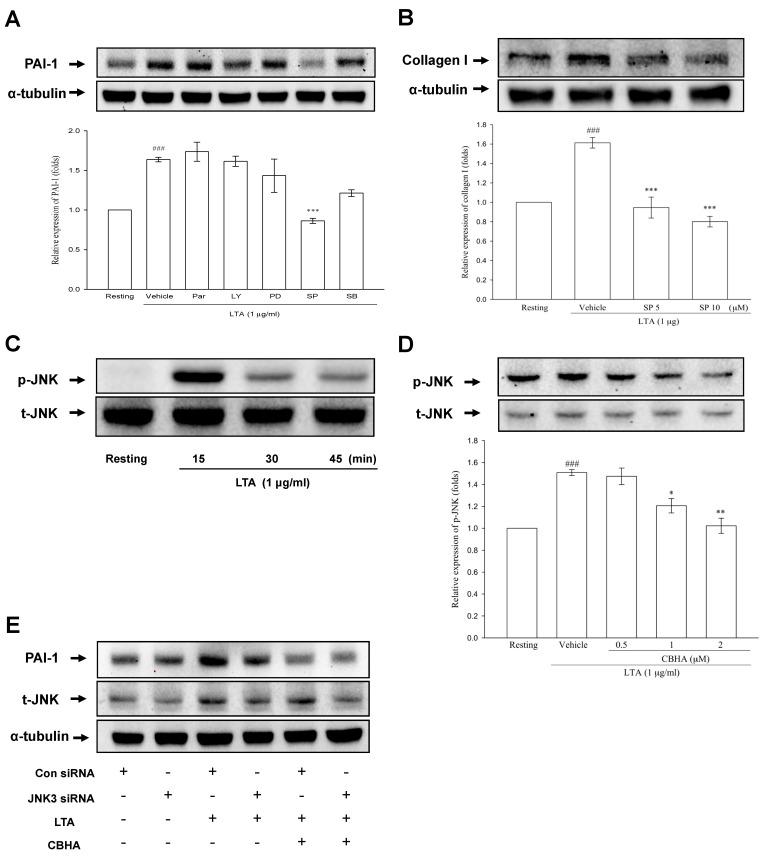
CBHA blocks LTA-induced JNK phosphorylation in human PMCs. (**A**) MeT-5A cells were treated with vehicle or various signal pathway inhibitors inclusive of parthenolide (Par, 10 μM), LY294002 (LY, 20 μM), PD98059 (PD, 20 μM), SB203580 (SB, 10 μM) and SP600125 (SP, 10 μM), respectively, then stimulated with LTA for 24 h. PAI-1 expression was analyzed by western blot and measured by optical densitometry analysis. Data were pooled from three independent experiments and expressed as average fold change (mean ± SEM) in the expression relative to resting or vehicle group. (**B**) MeT-5A cells pretreated with vehicle or SP600125 (SP, 5‒10 μM) were stimulated with LTA for 24 h. Collagen I expression was analyzed by western blot and measured by optical densitometry analysis. Data were pooled from three independent experiments and expressed as average fold change (mean ± SEM) in the expression relative to resting or vehicle group. (**C**) MeT-5A cells were treated with LTA for the indicated times, and JNK phosphorylation was evaluated. (**D**) MeT-5A cells were treated with LTA, with or without CBHA pretreatment (0.5, 1 or 2 μM), and the expression levels of phosphorylated or total JNK were examined in cell lysates by western blot and measured by optical densitometry analysis. Data were pooled from three independent experiments and expressed as average fold change (mean ± SEM) in the expression relative to resting or vehicle group. (**E**) MeT-5A cells transfected with control siRNA or JNK3 siRNA were stimulated by LTA for 24 h with or without CBHA pretreatment (1 μM), and the expression of PAI-1 and total JNK were examined by western blot. ^###^
*p* < 0.001 compared with the resting group; * *p* < 0.05, ** *p* < 0.01, *** *p* < 0.001 compared with the vehicle (DMSO) group. JNK, c-Jun N-terminal kinases, siRNA, small interfering RNA.

**Figure 4 pharmaceuticals-14-00585-f004:**
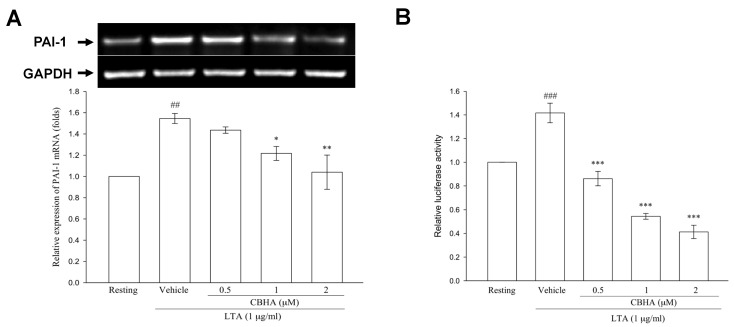
CBHA inhibits LTA-induced PAI-1 gene transcription in human PMCs. (**A**) MeT-5A cells were pretreated with the vehicle or the indicated concentrations of CBHA (0.5, 1 or 2 μM), and then stimulated with LTA for 6 h. PAI-1 mRNA level was detected by RT-PCR analysis. Data were pooled from three independent experiments and expressed as average fold change (mean ± SEM) in the expression relative to resting or vehicle group. (**B**) MeT-5A cells were transfected with both the specific PAI-1 reporter plasmid (p800Luc) and internal plasmid (Renilla), then treated with different concentrations of CBHA, and followed by LTA stimulation for 24 h. The luciferase activity of PAI-1 reporter gene was assessed. Data were pooled from three independent experiments and expressed as average fold change (mean ± SEM) in the activity relative to resting or vehicle group. ^##^
*p* < 0.01, ^###^
*p* < 0.001 compared with the resting group; * *p* < 0.05, ** *p* < 0.01, *** *p* < 0.001 compared with the vehicle (DMSO) group.

**Figure 5 pharmaceuticals-14-00585-f005:**
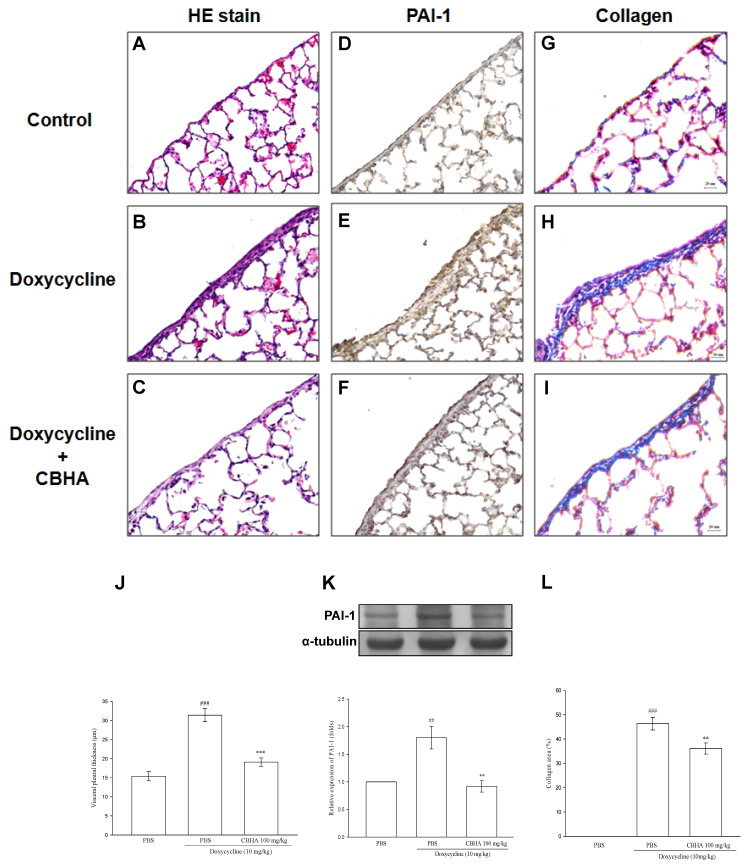
CBHA attenuates doxycycline-induced pleural thickening, PAI-1 expression and collagen deposition in viscera pleurae of rats. Photomicrographs of visceral pleura from haematoxylin and eosin (HE)-stained (**A**–**C**), PAI-1 immunostained (**D**–**F**) and Masson trichrome-stained (collagen, **G**–**I**) paraffin sections of pleura tissue obtained at autopsy 7 days after intrapleural injection with PBS (control), doxycycline (10 mg/Kg) plus PBS, or doxycycline plus CBHA (100 mg/Kg). (**J**) Histomorphometric quantification of visceral pleural thickness on pleural tissue resected 7 days after intrapleural injection of indicated agents. Ten random measures per section per animal (*n* = 3 for each group) were obtained. (**K**) PAI-1 expression in pleural tissue homogenates was analyzed by western blot and measured by optical densitometry analysis. Data were pooled from four experimental rat groups (*n* = 4 for each group) and expressed as average fold change (mean ± SEM) in the expression relative to PBS or doxycycline plus PBS group. (**L**) Quantitative percent of collagen-positive area in visceral pleura. Ten random measures per section per animal (*n* = 3 for each group) were obtained. Scale bar = 20 μm. ^##^ *p* < 0.01, ^###^ *p* < 0.001 compared with the PBS (control) group; ** *p* < 0.01, *** *p* < 0.001 compared with the doxycycline plus PBS group. PBS, phosphate buffered saline.

**Figure 6 pharmaceuticals-14-00585-f006:**
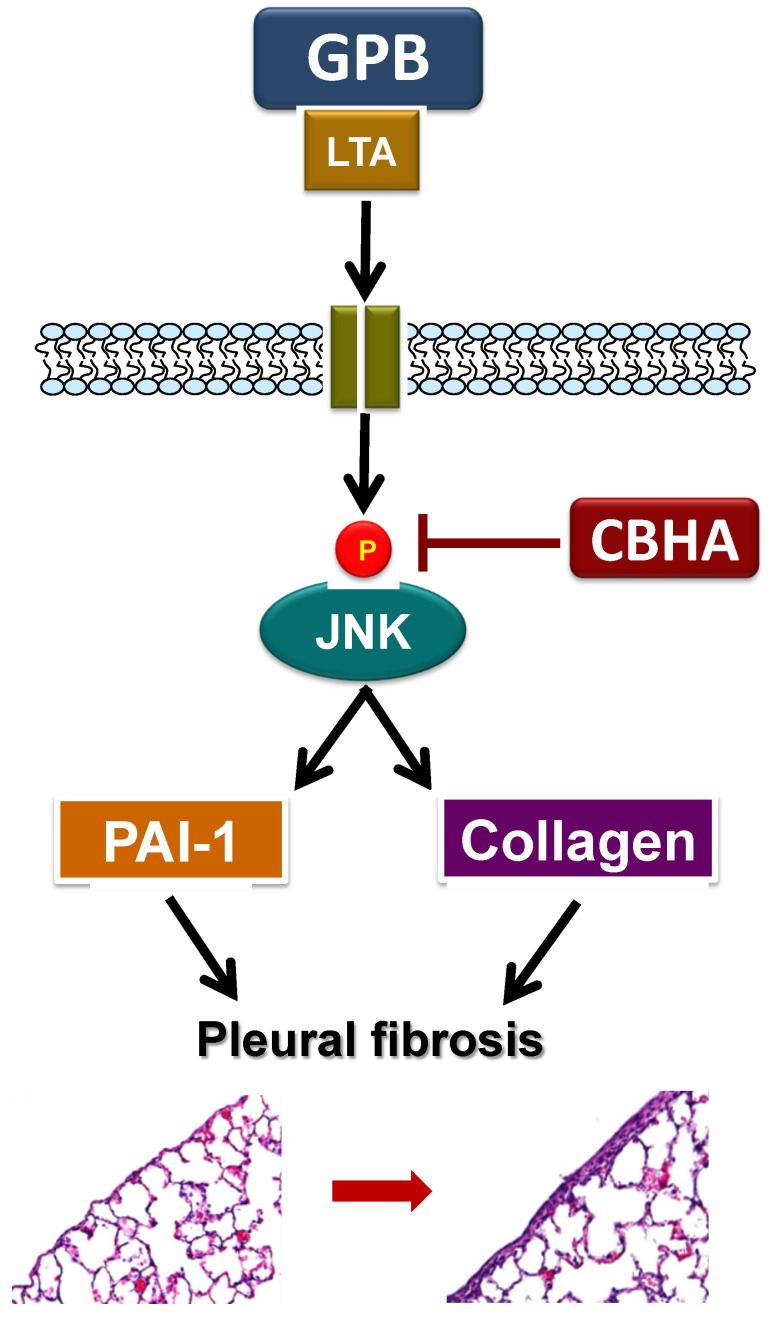
Graphic abstract illustrates that CBHA ablates LTA–induced PAI-1 and collagen production in human pleural mesothelial cells via down-regulating the JNK pathway, and attenuates pleural fibrosis (see text for more explications). Single arrow, established pathway. GPB, Gram-positive bacteria; LTA, lipoteichoic acid; CBHA, m-carboxycinnamic acid bis-hydroxamide; JNK, c-Jun N-terminal kinases; Ⓟ, phosphorylated.

**Table 1 pharmaceuticals-14-00585-t001:** Clinical and pleural effusion characteristics, and levels of fibrinolytic factors and cytokines in pleural fluid among all patients (*n* = 64) ^†^.

Characteristic	GPB PPE		*p*-Value *
	RPT ≤ 10 mm	RPT > 10 mm	
Subject, *n*	38	26	
Age, years	70 (42–80)	69 (48–78)	0.472
Males, *n*	25	17	0.973
Symptom onset to enrollment, days	8 (5–11)	9 (6–12)	0.604
Pleural fluid			
pH value	7.31 (6.45–7.50)	7.10 (6.42–7.39)	0.126
Glucose, mg/dL	98 (15–291)	63 (16–185)	0.570
Protein, g/L	4.3 (2.7–7.5)	4.7 (3.2–7.5)	0.193
LDH, IU/dL	599 (258–1760)	907 (263–5190)	0.019
Leukocyte count, cells/μL	3565 (1050–23,460)	8865 (1170–33,250)	0.058
PAI-1, ng/mL	57.5 (16.7–90.0)	99.0 (72.9–140.3)	<0.001
tPA, ng/mL	14.9 (3.2–21.0)	12.8 (3.1–18.3)	0.071
TNF-α, pg/mL	31.9 (21.8–85.9)	42.6 (23.7.8–110.3)	0.026
IL-1β, pg/mL	34.3 (21.0–94.5)	50.3 (23.0–101.6)	0.042
IL-8, pg/mL	590.6 (35.7–1906.2)	773.4 (121.7–1572.0)	0.036
Pleural fluid bacteria culture			0.595
*Streptococcus* spp., *n*	15	12	
*Staphylococcus* spp., *n*	23	14	
Initial pleural effusion CXR score, %	48 (25–65)	54 (32–81)	0.017
Residual pleural shadowing CXR score, %	1.9 (0.5–4.2)	4.5 (3.1–8.2)	<0.001
FVC at 12 months, % predicted	80 (76–84)	73 (70–76)	<0.001

Definition of abbreviations: GPB, gram-positive bacteria; PPE, parapneumonic pleural effusion; RPT, residual pleural thickening; LDH, lactate dehydrogenase; PAI-1, plasminogen activator inhibitor-1; t-PA, tissue-type plasminogen activator; TNF-α, tumor necrosis factor-α; IL-1β, interleukin-1β; IL-8, interleukin-8; spp., species; CXR, chest radiograph; FVC, forced vital capacity. ^†^ Data expressed as median (range). * Comparison between RPT ≤ 10 mm and RPT > 10 mm groups.

**Table 2 pharmaceuticals-14-00585-t002:** Multivariate logistic regression analyses of variables related to RPT > 10 mm among GPB PPE patients (*n* = 64).

Variables	OR	95% CI	*p*-Value
LDH, IU/dL	1.00	0.99‒1.00	0.135
PAI-1, ng/mL	2.72	1.27‒5.84	0.006
TNF-α, pg/mL	1.08	1.01‒1.23	0.055
IL-1β, pg/mL	1.05	0.99‒1.15	0.272
IL-8, pg/mL	1.00	0.99‒1.01	0.989

Definition of abbreviations: OR, odds ratio; CI, confidence interval.

## Data Availability

The data presented in this study are available on request from the corresponding author.

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
