# Peer review of "HDAC Inhibitor Abrogates LTA−Induced PAI-1 Expression in Pleural Mesothelial Cells and Attenuates Experimental Pleural Fibrosis"

_pharmaceuticals, 2021, doi:10.3390/ph14060585_

Round 1
Reviewer 1 Report
Authors performed the requested experiments showing a partial involvement of JNK activation in their system. Authors should still change figure legends describing data obtained from Western Blot by Optical densitometry analysis. Data shown in the figures are fold change and not only means of different experiments.
Reviewer 2 Report
Thank you for the new version of the manuscript; all my comments were taken into account by the authors. I have some minor points and remarks.
-Page 4, line 35, I think it should be better to slightly modify the sentence to add the information on what kind of predictive value is PAI-1. The PAI-1 level presents an excellent predictive value on RPT>10mm if I understand the paper well.
- Page 4 lin 36 “Correlate well with….” Maybe splitting this long and complex sentence would help the reader accurately understand the sentence's information.
-Page 5, line 46 “were stimulated with LTA” better suited.
- Page 5. The maximal effect is recorded with 1mg/ml of LTA. Do authors have some explanation or hypothesis on why dose-response is not observed?
-Page 5, line 47, I think that the sentence must be rewritten in that way: LTA stimulates the production of PAI-1” No data on the expression level of PAI-1 mRNA are given.
- Line 51, “Abrogate the stimulatory effect of LTA on PAI-1 production”.
